# Positive Roles of Resveratrol in Early Development of Testicular Germ Cells against Maternal Restraint Stress in Mice

**DOI:** 10.3390/ani10010122

**Published:** 2020-01-12

**Authors:** Sheeraz Mustafa, Wael Ennab, Korejo Nazar, Quanwei Wei, Zengpeng Lv, Zhicheng Shi, Fangxiong Shi

**Affiliations:** 1College of Animal Science and Technology, Nanjing Agricultural University, Nanjing 210095, China; sheerazmustafa786@gmail.com (S.M.); 2016105109@njau.edu.cn (W.E.); weiquanwei@njau.edu.cn (Q.W.); lvzengpeng@njau.edu.cn (Z.L.); 2017105023@njau.edu.cn (Z.S.); 2Faculty of Animal Husbandry and Veterinary Science, Sindh Agriculture University Tando Jam, Tando Jam 70060, Pakistan; nakorejo@sau.edu.pk

**Keywords:** resveratrol, maternal restraint stress, early development, testis

## Abstract

**Simple Summary:**

Maternal stress during pregnancy affected the early programming of the brain in the fetus with changes in neuroendocrine regulation, and offspring behavior was proven in the literature. In addition, resveratrol (RES) was reported to play positive roles against stress. However, how maternal stress affects testicular development and whether RES has potential protecting roles is unknown.

**Abstract:**

Our present study was designed to evaluate the effects of resveratrol (RES) in Swiss mice by exposing them to prenatal stress. Twenty-four Swiss mice were divided into four groups: control (C), maternal restraint stress (MRS), maternal restraint stress + resveratrol (MRS + RES) 2 mg, and maternal restraint stress + resveratrol (MRS + RES) 20 mg. Dams were exposed to stress by restraint in plastic tubes for four hours a day from 12–18 days of gestation. The results showed that male pups of MRS were significantly decreased in the testis weight, anogenital distance, area of seminiferous tubules, diameter of seminiferous tubules, area of the lumen, diameter of the lumen, and epithelial height of seminiferous tubules. However, the anomalies of the reproductive tract produced under restraint stress were neutralized by the use of RES 2 mg/kg. A significant difference was observed between terminal deoxynucleotidyl transferase dUTP nick end labeling (TUNEL)- positive germ cells in MRS and MRS + RES 20 mg/kg groups, while it was non-significant between MRS + RES 2 mg/kg and C groups. Apart from these effects, blood glucose levels were increased in MRS and MRS + RES 20 mg/kg groups, while experimental animals of the MRS + RES 2 mg/kg group significantly recovered. These results suggested that a lower dose of RES could cure the adverse effects of prenatal stress in early age male progeny. Thus, our study suggests, for the first time, practical values for a lower dose of RES 2 mg/kg as a safe and effective agent in the first week age of prenatally stressed mice.

## 1. Introduction 

The life of a developing body in the womb is extremely weak against environmental challenges [1,2]. Such harmful effects to a newborn can be from the mother’s malnutrition, exogenous glucocorticoids, and maternal stress [3]. During pregnancy and postnatal periods, maternal stress is common. Stress disorders are quite common in the newborn, affecting about 5–18% [4,5] and these include decreased anogenital distance, delayed testicular reduction [6], increase in the terminal deoxynucleotidyl transferase (dUTP) nick end labeling (TUNEL) and active caspase-3 neurons (apoptotic index) in the testis [7], reduction in body weight gain [8] and the effect on the developmental ability of offspring gametes by causing oxidative stress with the use of antioxidants with in vitro or in vivo experiments, in which the phenomenon was changed [9]. The main function of the testis is the production of sperm and androgens, which relies on the normal development of both testicular somatic cells and germ cells [10]. This process takes place in the rat between 13.5 and 14.5 post-conception days (pcd) and in the mouse between 11.5 and 12.5 pcd. Instead, until day 16 postpartum in the rat and day 17 postpartum in the mouse, when the Sertoli cell population is certainly established, Sertoli cells begin to proliferate [11]. The development of testosterone is detected in the mouse from 12.5 pcd [12]. However, it is still uncertain whether maternal restraint stress (MRS) will cause oxidative stress in testis tissues of neonatal mice. Indeed, significant proof from clinical and animal experiments indicates that stress reliably alters the metabolism of glucose, resulting in hyperglycemia, and plays a part in the induction of insulin resistance in various tissues [13,14]. However, there are limited data available about the role of MRS and resulting hyperglycemia in modified etiology and diabetes in neonates. 

In China, roughly 37.1% of mothers felt anxiety and depression [15]. Traditional medicine of China (TMC) is commonly used as an herbal medicine with 2000 years of history worldwide. Efficacy and clearly demonstrated mechanism theories from dietary plants and herbs are widely used in alternative medicines. Because of the various therapeutic capabilities, active compounds isolated from herbs and plants are well described [16,17]. One of the effective compounds from TMC is resveratrol (RES), which has beneficial effects of supplementation in humans and animals that are widely studied, but studies during pregnancy on animal metabolic health are limited. Furthermore, several studies have shown that oral RES is well absorbed and metabolized rapidly without significant toxicity [18,19]. Found with good concentration in the grape’s skin and seed, it occurs in a trans or a cis isoform [20] with biological effects such as an antioxidant via prevention of lipid peroxidation [21], neuronal cell death reduction [22], and reduction in traumatic brain injury [23]. It is still unknown whether the apoptosis of testis germ cells including neuronal damage in offspring is also due to increased MRS. 

The main aim behind the research was to cure the increased MRS of the fetus, which consequently enhances the maternal corticosterone, the main glucocorticoid in the mice. This exposure programs the irreversible cellular changes in the fetus, resulting in decreased birth weight and long-term changes in the adult offspring’s glucose metabolism and testis tissues. Thus, we hypothesized that MRS may cause testis damage through neuronal injuries with the MRS mechanism. However, the reproductive impairment may be dependent on its sensitivity or vulnerability to MRS. The RES can cross the blood–brain barrier, exerts potent antioxidant features [24], and can cross the placenta barrier with no adverse reproductive effects and mortality in embryo-fetal toxicity [25]. Previous research using a genetic mouse model of gestational diabetes mellitus showed that RES consumption (10 mg/kg body weight per day) considerably reduced hyperglycemia, enhanced insulin resistance, enhanced fetal survival, and reduced body weight at birth with improved and reduced activation of adenosine monophosphate-activated protein kinase (AMPK) in pregnant mice and offspring [26]. Evidence in previous supporting studies shows the neuroprotective and the antioxidant activities of RES in adult animals. To the best of our knowledge, the efficacy of RES against MRS has not been well studied. The specific aim of this study was to determine the curative effects of RES on MRS administered during the third week of pregnancy on birth weight and inspection of male reproduction in the first week of the offspring’s age.

## 2. Materials and Methods

### 2.1. Animals and Treatments 

The mouse still has more value because it reflects human and other animal development and disease for the genetic improvement, and it is fast and easy to approach in the area of research. We obtained adult male and female Swiss mice (28–32 g) from the Qinglongshan Laboratory Animal Company (Nanjing, China) by young male and female Swiss ICR (Institute for Cancer Research). Animals were quarantined for 7 days. With the ratio of (2:1), females were mated with males until copulation was detected. The day on which a vaginal plug was found was considered as day 0 of gestation. The present study induced stress from gestational (G) days 12 to 18, representing a period thought to cover a large extent of the human second trimester [27,28]; additionally, rats have been shown to be particularly susceptible to environmental influences, inflammatory processes, and stress [29]. The control group was left undisturbed in their home cages. The experimental protocols for mice were in accordance with the Guide for the Care and Use of Laboratory Animals organized by Nanjing Agricultural University Authorization Committee for Institutional Animal Care and Use (approval numbers: 31572403 and 31402075).

Animals were grouped as follows:Group 1: control (C) pups belonging to the normal pregnant mice;Group 2: maternal restraint stress (MRS) pups associated with pregnant mice who received restraint stress from G 12 to G 18;Group 3: (MRS + 2 mg) pups belonging to pregnant mice who received resveratrol at a dose of 2 mg/kg body weight (oral) from G 12 to G 18;Group 4: (MRS + 20 mg) pups associated with pregnant mice who received resveratrol at a dose of 20 mg/kg body weight (oral) from G 12 to G 18.

### 2.2. Experimental Procedures

Male offspring from each litter were sacrificed using chloroform vapor at the age of 7 days post-partum, and body weights were recorded. The testes were dissected, weighed, and fixed in 4% paraformaldehyde. The transverse sections 5 μm thick were cut from the middle part of each testis and stained by hematoxylin and eosin (HE). The standardized size (8 pups) on postnatal day (PND) 1 was another possible factor added to avoid the wide range of litter size for postnatal development. Dams of other groups were allowed to raise their entire litters [30,31,32] to avoid the effect of maternal lactation by culling litters.

### 2.3. Testicular Tissue Sampling

Testes were removed, weighed, and washed with 0.9% saline solution. Paraformaldehyde (4%, Sigma-Aldrich, St. Louis, MO, USA, purity 95%) was perfused (volume 0.5 mL, flux 0.1 mL/s) through the testicular artery. Testes were processed and paraffin-embedded (Paraplast-plus, McCormick Scientific, Passaic, NJ, USA). Transverse sections (5 μm) from the middle region of testes of each animal were cut from blocks using a microtome (Leica, Vienna, Austria) and mounted on poly-L-lysine hydrobromide (high purity grade, Sigma Aldrich, USA) treated slides.

### 2.4. Drugs

Resveratrol (Sigma Chemical Co., St Louis, MO, USA) was diluted to 5% in the vehicle (CMC, sodium carboxymethyl cellulose) [33,34]. The Beijing Institute for Pharmacology and Toxicology (China) supplied sodium carboxymethyl cellulose with trans-resveratrol for oral gavage at 0.5% in CMC.

### 2.5. TUNEL Staining

The apoptotic index was calculated as the percentage of cells that tested TUNEL-positive with the Colorimetric TUNEL Apoptosis Assay Kit to follow our previous laboratory work [35]. The paraffin portion of the experiment was initially deparafinized, rehydrated, and then incubated for 20 min at room temperature with Proteinase K (20 g/mL). The sections were incubated at 37 °C for 60 min in the dark with the Terminal Deoxynucleotidyl Transferase (TdT) enzyme buffer containing double diluted H2O, equilibration buffer, Bright Green Labeling Mix, and recombinant TdT enzyme. Eventually, 40 6-diamidino-2-phenylindole staining solution (C1005, Beyotime Biotechnology, Shanghai, China) stained the parts for 5 min in the dark. The negative control was conducted as above, but the TdT enzyme buffer was not incubated to ensure that no non-specific reactions occurred in the experiment.

### 2.6. Statistical Analysis

For the analysis of statistics, Graph Pad Prism 7 was used. All data are presented as standard error of the mean [36]. All areas and diameters were measured on the basis of geometrical constant “Pi” square root (A = π √2) (A = 3.14 √2) [33]. In order to compare histopathological parameters and MRS values, one-way analysis of variance [37] results for multiple comparison tests by Dennett’s multiple comparisons (which compares the mean of each column with the mean of control column) were carried out. The significant level of statistics was resolved at *p* < 0.05.

## 3. Results

During the experiment, interactions between control, MRS, MRS + 2 mg/kg, and MRS + 20 mg/kg groups were identified. Thus, only the main effects of the measurements and the growth performance in all groups were reported, i.e., area of seminiferous tubule, diameter of seminiferous tubule, area and diameter of lumen, epithelial height of seminiferous tubules, blood glucose level, average number of TUNEL-positive germ cells, and fetal testis growth.

### 3.1. Quantitative Histologic Evaluations of Postnatal Testis

In our findings during quantitative histological measurements (Figure 1i,ii), area of seminiferous tubules was significantly decreased in the groups of MRS and MRS + 20 mg/kg in comparison with the control group, while no statistical difference was seen in the MRS + 2 mg/kg group. The diameter of seminiferous tubules in MRS and MRS + 20 was significantly decreased, while this was non-significant in MRS + 2 mg/kg compared with the control (Figure 1A,B). The values for the area of lumen in MRS, MRS + 2 mg/kg, and MRS + 20 mg/kg and the diameter of lumen in MRS, MRS + 2 mg/kg, and MRS + 20 mg/kg were non-significantly different from the control group. Results are described as SEM, as mentioned in Figure 1C,D. The epithelial heights in the groups of MRS and MRS + 20 mg/kg were more significant than those of MRS + 2 mg/kg (Figure 1E). 

### 3.2. Blood Glucose Levels 

The high values of blood glucose (5.0 mmol/L) were seen in the samples from the MRS and the MRS + 20 mg/kg groups (4.9 mmol/L). With the help of sinocare blood glucose monitor, blood glucose levels were noted during the study period. Results in the group of MRS = (0.0004) and in the group of MRS + 20 mg/kg = (0.0001), thus crossing the limit at (*p* < 0.05), and the presented blood glucose levels were significantly increased, while they were non-significant in the MRS + 2 mg/kg = (0.2636) group in comparison to the control group (Figure 2A).

### 3.3. Anogenital Distance

The significance levels of anogenital distance in MRS = (0.0028) and MRS + 20 mg/kg = (0.0011) were different, while data showed that distance was improved in MRS + 2 mg/kg = (0.0513) (Figure 2B).

### 3.4. Testis Weights

Significant reductions in the testis weight of MRS = (0.0013) and MRS + 20 mg/kg = (0.0001) of 7 days old pups were observed during the study, while in MRS + 2 mg/kg = (0.0792), findings were non-significant in comparison to the control (Figure 2C).

### 3.5. Average Number of TUNEL-Positive Germ Cells/Tubule

Percentage of seminiferous tubules with TUNEL-positive testicular germ cells was more significantly apoptotic in the MRS = (0.0001) and the MRS + 20 mg/kg = (0.0001) groups as compared to the MRS + 2 mg/kg = (0.0014) group, although the MRS + 2 mg/kg group pups’ germ cells were also significantly different (*p* < 0.05) compared to the control group pups’ germ cells (Figure 2D and Figure 3).

## 4. Discussion 

Throughout the ages, infertility issues have been a significant health problem; 8–12% of couples are affected worldwide, and about 40–50% of these are due to infertility of the “male”, as 2% of all males have sperm parameters among all infertility cases [38]. Experimental animal studies indicate that prolonged stress increases the risk of many health problems, including neuropsychiatric disorders [39], type 2 diabetes [40], and tumorigenesis [41]. It has been reported that maternal stress, such as stress during pregnancy [42], could alter body weight and glucose metabolism in offspring. However, it is unclear whether paternal stress could affect glucose metabolism in their offspring. It is important to understand mechanisms such as altered placental activity, which allows more stress hormone cortisol to move through the fetus [43] and affects the essential role of the maternal immune system [44]. Faster physical development as well as more anxiety can be caused by prenatal stress in the neonate [45]. Here, we provided shreds of evidence that MRS across generations of timed pregnant mice has downstream impacts on germ cell development and testicular weight. In our findings, there were no significant interactions between control and MRS = 2 mg/kg for the TUNEL-positive germ cells and the testicular weight. The reported study suggests that an increase in the percentage of TUNEL and active caspase-3 positive cells (apoptotic index) in the testis was observed, which enhanced testicular cell death and reduced testosterone concentrations, sperm quality, and fertility in prenatally stressed males. They also proved that the low testicular weight found in MRS group males could be explained by reduced spermatogenesis due to a greater apoptosis index found in germ cells [7,46].

It was suggested that the overall effect of stress on glucose levels results from reductions in gestational lengths accompanied by altered blood glucose concentrations in late pregnancy and postpartum periods, where ancestral exposure to epigenetic stress programs predetermine the risk of birth and adverse effects on mothers and newborns [47]. Our results of blood glucose levels were observed to be significantly increased in the MRS and the MRS + 20 mg/kg groups, while no significant difference was found in the group of MRS + 2 mg/kg compared with the control. This suggests that 2 mg/kg RES had positive roles. Our results show that the anogenital length was lower in the group of MRS than in the control group, though it was notably non-significantly decreased in the group of MRS + 2 mg/kg. With these results, the F1 generation became evident by postnatal day 7. Likewise, in previous findings, prenatal stress was also associated with decreased anogenital distance accompanied by lower plasma levels of luteinizing hormone (LH), follicular stimulating hormone (FSH), testosterone, and delayed testicular reduction [6]. LH is able to stimulate testosterone synthesis and secretion from rat tests as young as embryonic day 14.5 [48]. Recently, Gross et al. [49] observed that prenatal stress can increase maternal glucocorticoid concentrations, which limit fetal growth with variable effects on postnatal development—even glucocorticoid receptors and 11b-hydroxysteroid dehydrogenase-2 are key regulators of stress hormones and the enzymes that inactivate glucocorticoid. Furthermore, Dey and his team [50] cited that developmental directions of offspring were heeled as early as postnatal day 7 with the lower dose of RES 2 mg/kg. RES was found to have a protective effect at a lower dose (2 mg/kg) and a contraindicative effect at a higher dose (20 mg/kg). The results suggest that the mechanisms in the gestational period can be involved through the maternal blood.

In quantitative histological evaluations of fetal testis, the area and the diameter of seminiferous tubules and epithelial height were significantly decreased in MRS and MRS + 20 mg/kg groups, though they were observed to be non-significant in the MRS + 2 mg/kg group in comparison to the control group. The area and the diameter of the lumen were significantly decreased in MRS, MRS + 2 mg/kg and MRS + 20 mg/kg groups compared to the control group. We hypothesized that the decrease in the tubules was due to a decreased growth rate during maternal stress followed by a stress-induced and consequent release of placental corticotropin-releasing hormone (CRH) to the intrauterine environment [51,52,53]. In previous studies, it was suggested that MRS affects the ergic system of gamma aminobutyric acid (GABA) by altering the expression of Na-K-Cl cotransporter 1 (NKCC1), potassium chloride cotransporter 2 (KCC2), and gamma-Aminobutyric acid (GABAA) receptora1 and a5 in a rat pup hippocampus, and changes showed the dysregulation of inhibitory neurotransmission in early life [54]. 

Investigations in the hypogonadal mouse, which lacks the gonadotropin-releasing hormone (GnRH) peptide, also demonstrate that GnRH is a critical regulator of pituitary and gonadal function around the time of birth in mice [55]. Furthermore, newborns who do not achieve their constitutional development capacity in utero because of maternal psychological stress encounter fetal growth restrictions [56,57,58], which has a number of consequences for the developmental line of offspring. Another study also suggested that the key regulators are stress hormones for prenatal stress, which may boost maternal glucocorticoid levels that limit fetal growth with variable effects on postnatal growth [49,59]. The main finding of this study is the curative ability of cellular death with the lower dose of RES 2 mg/kg, which is influenced by MRS and might lead to signals that occur through stress hormones to the fetus, although it has not been dealt with yet.

## 5. Conclusions

Psychological stress in offspring encourages hepatic gluconeogenesis and hyperglycemia with elevated parental glucocorticoids [60]. Results from this research show that the lower dose of RES 2 mg/kg in MRS is associated with a lower risk of cell death, lower testicular weight, and decreased blood glucose levels, while a higher dose did not show any satisfactory results; thus, we suggest that the usage of high dose RES during the gestation period may lead to unfavorable conditions for a newborn in comparison to a lower dose.

## Figures and Tables

**Figure 1 animals-10-00122-f001:**
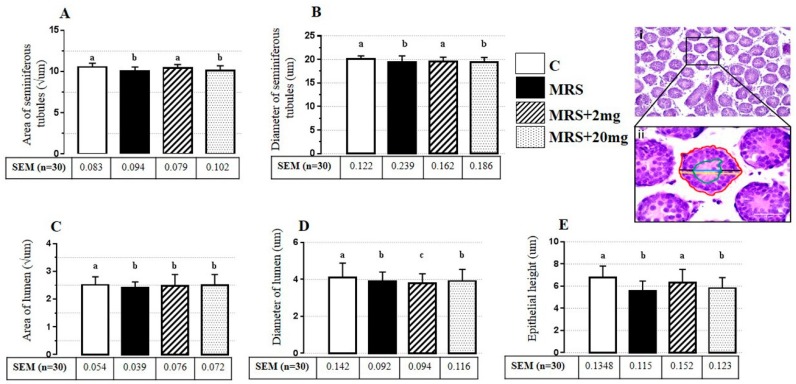
Photomicrograph of one week old mouse testis showing histologic figure (**i**) and measurements (**ii**). Scale bar 10 µm. Area of seminiferous tubules (red), diameter of seminiferous tubules (yellow), area of the lumen (green), diameter of the lumen (blue), epithelial height (black). Histologic measurements of 7 days old mice pups. (**A**) The area and (**B**) the diameter of seminiferous tubules significantly decreased in the groups of maternal restraint stress (MRS) and 20 mg/kg, though they recovered in the 2 mg/kg group in both graph pictures, while there was no curative significant effect of 2 mg/kg on (**C**) the area and (**D**) the diameter of the lumen, although they decreased in the MRS and the 20 mg/kg groups. (**E**) The epithelial height of the seminiferous tubules was decreased in the groups of maternal restraint stress and 20 mg/kg, while it increased in the 2 mg/kg group. Alphabets indicate significance: *p* < 0.05 compared to the non-stress controls.

**Figure 2 animals-10-00122-f002:**
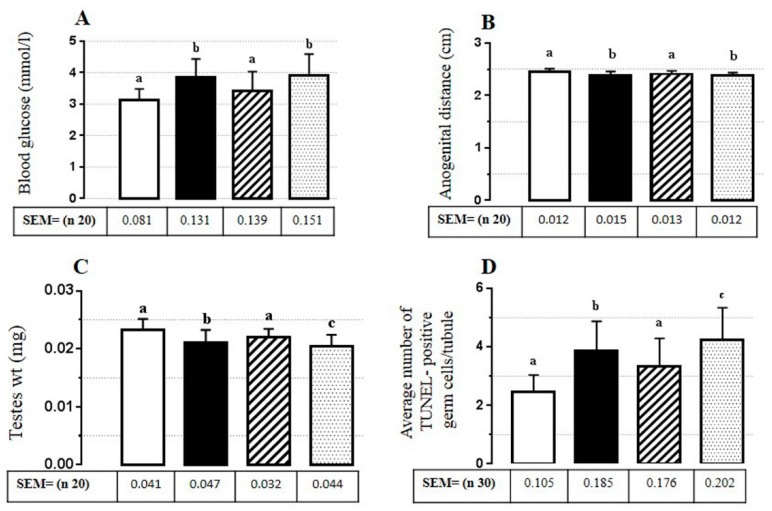
Blood glucose level, anogenital distance, testis weight, and terminal deoxynucleotidyl transferase dUTP nick end labeling (TUNEL)-positive germ cells in experimental mice. (**A**) The dose of 2 mg/kg resveratrol (RES) in maternal restraint stress decreased blood glucose level in pups 7 days of age. Glucose was mainly increased by maternal restraint stress and 20 mg/kg. (**B**) Resveratrol 2 mg also affected anogenital distance, which was mainly decreased in the groups of maternal restraint stress and 20 mg/kg dose. (**C**) The average number of TUNEL-positive cells significantly decreased in the group of 2 mg/kg and was significantly higher in the maternal restraint stress and the 20 mg/kg groups. (**D**) Testes weight decreased in maternal restraint stress and 20 mg/kg groups, while it increased in the group of 2 mg/kg. Alphabets indicate significance: *p* < 0.05 compared to the non-stress controls.

**Figure 3 animals-10-00122-f003:**
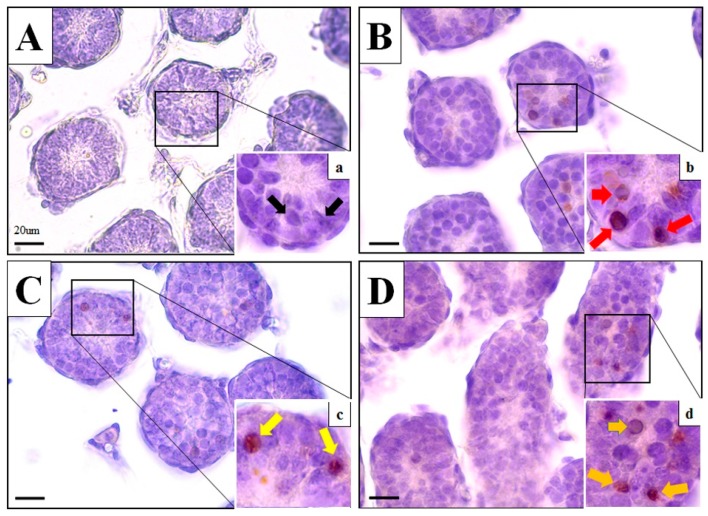
TUNEL-positive cells in testicular sections from control and maternal restraint stressed males. (**A**) Control testes; (**B**) MRS; (**C**) MRS + 2 mg/kg; and (**D**) MRS + 20 mg/kg seminiferous tubules. Magnification in panels (**A**–**D**) 400×; magnification in (a), (b), (c), and (d) 1000×.

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
