# Peer review of "Positive Roles of Resveratrol in Early Development of Testicular Germ Cells against Maternal Restraint Stress in Mice"

_animals, 2020, doi:10.3390/ani10010122_

Round 1
Reviewer 1 Report
The manuscript by Mustafa et al. investigated the positive roles of resveratrol in early development of testicular germ cells against maternal restraint stress in mice, which is very interesting, but the content is limited. Therefore, this manuscript could be considered for publication in Brief Research Report or Short Communication.
Additionally, some issues need to be addressed.
Please add some data for the process of the testicular development. Please add the method of TUNEL staining. Please add some pictures in Figure 1.
Author Response
Dear respectable.
Thank you for encouraging, after your valuable suggestions we have added material and some corrections accordingly in the response letter. Hopefully, our work will be encouraged further to publication.

Reviewer 2 Report
The conclusions would better fit with previous research if maternal glucocorticoid production had been measured; or another independent indicator of the varying degrees in (presumably) induced maternal stress.
Author Response
Dear respectable.
thank you for your valuable suggestion and encouragement to our work. we have taken seriously your suggestions and queries in a response letter to make our work more worthful. we hope our work will be accepted for publication.
